# Mechanisms and Therapeutic Potential of Multiple Forms of Cell Death in Myocardial Ischemia–Reperfusion Injury

**DOI:** 10.3390/ijms252413492

**Published:** 2024-12-17

**Authors:** Shinya Tsurusaki, Eddy Kizana

**Affiliations:** 1Centre for Heart Research, The Westmead Institute for Medical Research, The University of Sydney, Westmead, NSW 2145, Australia; shinya.tsurusaki@sydney.edu.au; 2Westmead Clinical School, The Faculty of Medicine and Health, The University of Sydney, Westmead, NSW 2145, Australia; 3Department of Cardiology, Westmead Hospital, Westmead, NSW 2145, Australia

**Keywords:** myocardial ischemia–reperfusion injury, apoptosis, necroptosis, ferroptosis, pyroptosis

## Abstract

Programmed cell death, especially programmed necrosis such as necroptosis, ferroptosis, and pyroptosis, has attracted significant attention recently. Traditionally, necrosis was thought to occur accidentally without signaling pathways, but recent discoveries have revealed that molecular pathways regulate certain forms of necrosis, similar to apoptosis. Accumulating evidence indicates that programmed necrosis is involved in the development of various diseases, including myocardial ischemia–reperfusion injury (MIRI). MIRI occurs when blood flow and oxygen return to an ischemic area, causing excessive production of reactive oxygen species. While this reperfusion is critical for treating myocardial infarction, it inevitably causes cellular damage via oxidative stress. Furthermore, this cellular damage triggers multiple forms of cardiomyocyte death, which is the primary cause of inflammation, cardiac tissue remodeling, and ensuing heart failure. Therefore, understanding the molecular mechanisms of various forms of cell death in MIRI is crucial for therapeutic target discovery. Developing therapeutic strategies to inhibit multiple cell death pathways simultaneously could provide effective protection against MIRI. In this paper, we review the fundamental molecular pathways and MIRI-specific mechanisms of apoptosis, necroptosis, ferroptosis, and pyroptosis. Additionally, we suggest that the simultaneous suppression of multiple cell death pathways could be an effective therapy and identify potential therapeutic targets for implementing this strategy.

## 1. Introduction

Myocardial infarction (MI) is a life-threatening event with high morbidity and mortality. MI is caused by the blockage of one or more coronary arteries due to atherosclerotic plaque rupture and the formation of blood clots, which disrupts blood flow and oxygen supply. The decrease in oxygen levels and prolonged hypoxia damage cardiomyocytes [1,2,3,4]. Therefore, the main strategy for MI treatment is to restore blood flow through interventions, such as stent insertion after balloon angioplasty or systemic thrombolytic therapy. However, due to the nature of these therapies, myocardial ischemia–reperfusion injury (MIRI) is an inevitable consequence during MI treatment, leading to further cardiomyocyte loss. Unlike other cell types in the heart, adult cardiomyocytes have limited regenerative capacity when injured [5]. Thus, extensive loss of cardiomyocytes results in inflammation, adverse cardiac remodeling, and cardiac dysfunction. Therefore, inhibiting cardiomyocyte death is important for protecting heart function from MIRI.

Cell death, which is induced by various factors such as homeostasis disruption, diseases, and physical stimuli, has traditionally been classified into two types, which are apoptosis and necrosis. While apoptosis, which is well-known as programmed cell death (PCD), occurs under the control of strict molecular pathways, necrosis was considered an uncontrolled cell death occurring accidentally without regulatory mechanisms. However, recent studies have revealed that our bodies harbor multiple forms of cell death. In particular, necroptosis, ferroptosis, and pyroptosis, which are newly identified types of necrosis regulated by molecular pathways, are now referred to as “programmed necrosis” and are included in PCDs. As apoptosis is executed by caspase-3/7 through the activation of intrinsic or extrinsic pathways, mixed lineage kinase domain-like (MLKL) and Gasdermin D (GSDMD) have the role of execution factors in necroptosis and pyroptosis, respectively, and the activation of execution factors is tightly regulated. While no specific execution factor has yet been identified for ferroptosis, its induction is triggered by the accumulation of lipid peroxides and a decline in antioxidant mechanisms.

The mechanisms and roles of PCDs are still being investigated intensively. In addition to apoptosis and necrosis, programmed necrosis is now thought to be involved in the onset and progression of a variety of diseases. In the case of MIRI, the number of studies reporting the contribution of programmed necrosis to MIRI has increased significantly over the past decade. This suggests that several types of cell death are induced in MIRI, either simultaneously or in stages. In fact, some therapeutic candidates with anti-apoptotic effects have been insufficient in protecting heart function from MIRI, highlighting the limitations of inhibiting only a single type of cell death [6,7]. Additionally, one study demonstrated that different patterns of cell death are induced between the right ventricle and the infarcted area of the left ventricle [8]. Furthermore, considering the history of necroptosis discovery, inhibiting a single type of cell death may potentially activate other forms of cell death through signaling pathway crosstalk. Therefore, in order to develop more effective treatment for MIRI, the simultaneous inhibition of multiple cell death pathways presents a promising strategy.

## 2. Various Forms of Programmed Cell Death

### 2.1. Apoptosis

Since the 1960s, it has been known that the spontaneous deletion of cells is a key factor in the developmental process, and this deletion was initially recognized as a type of necrosis called shrinkage necrosis. However, in 1972, Kerr et al. closely observed the characteristics of this shrinkage cell death and proposed the concept and term “apoptosis” as an inherently programmed event. In describing the morphology of apoptosis, they noted nuclear and cytoplasmic condensation, nuclear fragmentation, and the formation of small spherical cytoplasmic fragments (apoptotic bodies) that contain organelles. These morphological features are still used as the criteria for identifying apoptosis today [9].

The apoptosis pathway has been extensively investigated since its discovery and is broadly classified into intrinsic and extrinsic pathways. The intrinsic pathway is triggered by various factors such as DNA damage, endoplasmic reticulum (ER) stress, and oxidative stress. These stimuli upregulate the expression of bcl-2 homolog 3 (BH3)-only proteins, including BIM, BID, BMF, BAD, HRK, BIK, PUMA, and NOXA. BH3-only proteins bind to anti-apoptotic proteins in the B-cell lymphoma2 (BCL-2) family such as BCL-2, BCL-xl, and myeloid cell leukemia-1 (MCL-1) and work as their inhibitor; thus, pro-apoptotic proteins Bcl-2-associated X protein (BAX) and/or Bcl-2 homologous antagonist killer (BAK) are activated and form oligomers to generate pores in the outer mitochondrial membrane. Cytochrome C, which is a mitochondrial respiratory chain protein located in the mitochondrial intermembrane space, is released from these pores. In the cytosol, cytochrome C binds to apoptotic protease activating factor 1 (Apaf-1), leading to the formation of apoptosome. The apoptosome recruits and cleaves procaspase-9 into its active form, resulting in the sequential activation of caspase-3/7 and executing apoptosis [10].

On the other hand, the extrinsic pathway is activated by external stimuli such as death ligands including tumor necrosis factor α (TNFα) and Fas ligand. The binding of death ligands to death receptors such as the TNF receptor, Fas, and TRAIL receptor leads to the recruitment of procaspase-8 to the death receptor via Fas-associating protein with death domain (FADD) for caspase-8 activation by cleavage. Active caspase-8 cleaves procaspase-3 into its active form, and the cleaved caspse-3 acts as an executor of apoptosis, similar to the intrinsic pathway [11].

### 2.2. Necroptosis

Necroptosis was discovered as a form of necrotic cell death induced by TNFα, an extrinsic apoptosis inducer, in L929 cells [12]. Because the apoptosis inhibitor Z-VAD-FMK enhanced this unusual cell death in that study, it was initially thought that only particular cell types exhibit this phenotype. However, the general existence of necrosis with underlying molecular pathways was revealed due to the discovery of the necroptosis inhibitor Necrostatin-1 and its target receptor-interacting protein 1 (RIPK1) [13,14]. Following this, the other regulatory factors such as RIPK3 and mixed lineage kinase domain-like (MLKL) were also identified, and the molecular pathway has been gradually elucidated in two decades [15,16].

Necroptosis shares a part of its signaling pathway with extrinsic apoptosis. Normally, the stimuli to death receptors induce apoptosis as described above because active caspase-8 inhibits the necroptosis pathway through the cleavage of RIPK1 and RIPK3. On the other hand, in the case that caspase-8 is suppressed for any reason, RIPK1 and RIPK3 are activated through the phosphorylation of RIPK1 at Ser166 and RIPK3 at Ser227. Thr357 and Ser358 of MLKL are phosphorylated downstream of phosphorylated RIPK3. Phosphorylated MLKL (p-MLKL) forms an octamer consisting of two tetramers and ruptures the plasma membrane via pore formation [17]. As another signaling pathway, it was reported that RIPK3 activates Ca^2+^–calmodulin-dependent protein kinase (CaMKII) through the direct binding and phosphorylation of CaMKII at Thr287 when necroptosis is induced in cardiomyocytes with ischemia/reperfusion injury or doxorubicin-induced cardiac damage [18]. Additionally, accumulating evidence indicates that reactive oxygen species (ROS) production dependent on RIPK3 is related to necroptosis. In this case, ROS oxidizes CaMKII at Met281 and 282 residues, causing CaMKII activation, as well as phosphorylation at Thr287. CaMKII activation facilitates the opening of the mitochondrial permeability transition pore (mPTP) and depolarization of the mitochondrial membrane potential, resulting in necroptosis induction [18].

### 2.3. Ferroptosis

Ferroptosis is one of the necrotic forms of cell death that was identified in rat sarcoma (RAS) mutated cancer cells when treated with the chemotherapeutic drug erastin. These cancer cells showed caspase-independent, non-apoptotic cell death, and this cell death was inhibited by the chelation of iron; thus, it was named “ferroptosis”, derived from iron dependency. Additionally, ferroptosis is also known to depend on phospholipid peroxidation as lipophilic antioxidants like ferrostatin-1 can inhibit ferroptosis by eliminating lipid peroxides [19].

Some ferroptosis inducers including erastin target the System X_c_ transporter-related protein (xCT), which is a part of the cystine transporter known as SLC7A11 [20]. Since cystine is utilized for the biosynthesis of glutathione required for lipoperoxide reduction, the suppression of cystine uptake by SLC7A11 inhibition leads to a decrease in glutathione and ferroptosis induction via lipoperoxide accumulation. Additionally, glutathione peroxidase 4 (GPx4) has an important role in the ferroptosis pathway. While the GPx family has peroxidase activity to protect cells from oxidative damage by utilizing glutathione, only GPx4 preferentially reduces lipoperoxides as a substrate. Therefore, gene knockout and the chemical inhibition of GPx4 lead to the accumulation of lipoperoxides and ferroptosis induction [21]. Another key player in lipoperoxide reduction is ferroptosis suppressor protein 1 (FSP1). FSP1 recycles ubiquinol from ubiquinon in the presence of NAD(P)H after ubiquinol is utilized to eliminate lipoperoxides. This pathway provides an additional means for lipoperoxide reduction [22,23]. Furthermore, the Fenton reaction is largely related to ferroptosis induction. The Fenton reaction is a process that produces free radicals and rapidly amplifies lipoperoxides in a chain reaction using iron as a catalyst. Consequently, iron chelators inhibit ferroptosis by suppressing the Fenton reaction [24].

Regarding the substrate for lipid peroxidation, phosphatidylethanolamine (PE) containing arachidonic acid (AA) and adrenic acid (AdA) is identified as the most sensitive target. AA and AdA are selectively converted to AA-CoA and AdA-CoA by acyl-CoA synthetase long-chain family 4 (ACSL4), and this step strongly influences the abundance ratio of PE containing AA and AdA. ACSL4 knockout cells show decreased sensitivity to ferroptosis, indicating that ACSL4 is also a crucial factor in the ferroptosis pathway [25,26].

### 2.4. Pyroptosis

Pyroptosis, which is currently recognized as caspase-1/4/5/11-dependent necrotic cell death, was found in macrophages infected by Shigella flexneri [27]. This type of cell death was initially regarded as apoptosis due to caspase-1-dependent DNA double strand break and nuclear condensation. However, Cookson et al. renamed this process to pyroptosis because it causes inflammation by releasing intracellular components, while apoptosis is defined as a non-inflammatory programmed cell death [28]. The mechanisms of pyroptosis have been well investigated in immune cells; however, evidence in other cell types has been accumulating recently [29,30].

Pyroptosis is induced by the formation of the inflammasome complex. In the canonical pathway, the inflammasome typically consists of procaspase-1, apoptosis-associated speck-like protein containing a caspase recruitment domain (ASC), and NOD-like receptors (NLRs) such as NLRP1/3, and NLRC4. Stimulation by pathogen-associated molecular patterns (PAMPs) or damage-associated molecular patterns (DAMPs) leads to the activation and assembly of the inflammasome complex through toll-like receptors (TLRs) on the cell surface or through direct binding to NLRs in the cytosol. This results in the automatic cleavage of procaspase-1 into p14 and p20 subunits. Subsequently, active caspase-1 cleaves Gasdermin D (GSDMD), and the N-terminal fragment of digested GSDMD assembles into an octamer, forming pores in the plasma membrane as the execution factor. Additionally, pro-IL-1β and pro-IL-18 are also cleaved into their mature forms by active caspase-1 and released through the membrane pores formed by GSDMD [31].

Moreover, pyroptosis can also be induced by the activation of caspase-11 in rodents and caspase-4/5 in humans, known as the non-canonical pathway. The caspase activation and recruitment domain (CARD) of caspase-4/5/11 acts as a sensor for lipopolysaccharide (LPS). The binding of LPS to CARD causes the oligomerization and activation of caspase-4/5/11, unlike the canonical pathway, which involves the formation of an inflammatory complex with ASC and NLRs [32]. Active caspase-4/5/11 directly cleave GSDMD, leading to pyroptosis, whereas caspase-4/5/11 do not cleave pro-IL-1β and pro-IL-18. However, the activation of the caspase-1 inflammasome by caspase-4/5/11 leads to the maturation and release of IL-1β and IL-18 [31].

## 3. Involvement of Programmed Cell Death in Myocardial Ischemia–Reperfusion Injury

Cardiomyocyte loss occurs as a result of irreversible cell injury and death. It has been reported that necrosis and apoptosis are induced in myocardial ischemia/reperfusion injury (MIRI); however, recent evidence has shown that additional types of cell death are also involved (Figure 1). Therefore, we describe here the contributions and molecular mechanisms of apoptosis, necroptosis, ferroptosis, and pyroptosis in the progression of MIRI.

### 3.1. Apoptosis

The induction of myocardial apoptosis is triggered through various pathways. Endoplasmic reticulum (ER) stress impacts cardiac apoptosis. MIRI induces the elevation of ER stress markers such as GRP78 and phosphorylated PERK, which leads to an increase in CHOP expression, a well-known pro-apoptotic inducer in ER stress [33,34]. The upregulation of the PI3K/Akt signaling pathway is observed in MIRI models, while this pathway is downregulated in MI models, which can be attributed to the differences in injury models and the time course of analysis. However, further increases in the PI3K/Akt signaling pathway through pharmacological treatment and gene knockout such as SFRP4 are beneficial in suppressing cardiac apoptosis in MI models as well, suggesting that the PI3K/Akt signaling pathway is one of the key pathways for apoptosis induction in both MI and MIRI [35,36,37]. Nuclear factor kappa B (NF-kB), an important transcription factor for inflammatory responses, also contributes to apoptosis induction in MIRI. In oxygen–glucose deprivation/reoxygenation (OGD/R)-exposed cells, adipocyte enhancer-binding protein 1 (AEBP1), which has been shown to be involved in cardiac fibrosis in heart failure, is upregulated. AEBP1 suppresses the NF-kB inhibitory protein IkBα through direct binding, thereby inducing inflammatory responses and apoptosis via the activation of the NF-kB pathway. Knockdown of AEBP1 reduces inflammation and apoptosis in both in vitro and in vivo MIRI models [38]. Additionally, the extrinsic pathway also contributes to apoptosis in MIRI. After the induction of MIRI, the expression of Fas and the Fas ligand (FasL) is increased at both the mRNA and protein levels [39]. It has also been reported that an increase in FADD expression is found in MIRI models, and FADD knockout mice exhibit reduced apoptosis, infarct size, and cardiac dysfunction [40].

### 3.2. Necroptosis

Necroptosis has also been considered as a cause of cardiomyocyte loss in MIRI. Rat MIRI models and hypoxia/reoxygenation-injured cell models showed a significant increase in TNFα, RIPK1, RIPK3, and p-MLKL. Inhibition of the TNFα/RIPK1/RIPK3/MLKL pathway via resveratrol decreases the number of necroptotic cells and infarct size [41]. In addition to this regulatory role, RIPK3 is also involved in several necrotic pathways in the progression of MIRI. RIPK3 is known to form a complex with phosphoglycerate mutase 5 (PGAM5), which is a mitochondrial membrane protein functioning as a Ser/Thr phosphatase [42]. PGAM5 activates dynamin-related protein 1 (Drp1) independently of its phosphatase activity and activated Drp1 induces necroptosis through mitochondrial dysfunction. Therefore, the pharmacological inhibition or knockout of PGAM5 can protect the heart by suppressing necroptosis via the promotion of mitochondrial quality control [43,44]. In contrast, another study shows that PGAM5 protects cells from necroptosis through PINK1-mediated mitophagy, and PGAM5 deletion exacerbates rather than inhibits necroptosis in MIRI models [45]. Thus, the role of PGAM5 in the necroptosis pathway has been debated. The involvement of the PI3K/Akt pathway in necroptosis has been reported in MIRI as well as in apoptosis. A decrease in p-PI3K and p-Akt was observed following the downregulation of glucagon-like peptide-1 receptor (GLP-1R) after MIRI induction. Treatment with a GLP-1R agonist recovers the expression of the GLP-1R/PI3K/Akt pathway and inhibits necroptosis through the suppression of p-RIPK3 and p-MLKL [46]. Furthermore, studies using murine MIRI models and in vitro ischemia/reperfusion models have revealed that RIPK3 upregulation potentially induces ER stress and intracellular Ca^2+^ overload, leading to the activation of xanthine oxidase (XO) expression. Overproduction of reactive oxygen species (ROS) by XO mediates the mitochondrial permeability transition pore (mPTP) and necroptosis in cardiomyocytes. Therefore, ER stress inhibitors have the potential to improve heart function after MIRI [47]. Transient receptor potential canonical channel 6 (TRPC6) is also related to calcium regulation. TRPC6 is highly expressed in the MIRI group compared to controls, causing intracellular Ca^2+^ overload. An increase in Ca^2+^ concentration triggers the phosphorylation of CaMKII, which activates CaMKII and facilitates mPTP opening [48].

### 3.3. Ferroptosis

Recently, the involvement of ferroptosis in MIRI has been increasingly reported. Some articles indicate that a ferroptosis inducer ACSL4 is upregulated, while a suppressor GPx4 is downregulated in murine and in vitro MIRI models [49,50]. Furthermore, iron accumulation is also closely related to ferroptosis induction in MIRI. MIRI drives an increase in ubiquitin-specific protease 7 (USP7) and p53 expression; thus, transferrin receptor 1 (Tfr1), which controls iron uptake through binding to the transferrin–iron complex, is upregulated by the USP7/p53 pathway. This upregulation increases intracellular Fe^2+^ levels, causing lipid peroxidation [51]. In contrast, the protein level of ferritin heavy chain 1 (FTH1), which is a subunit of ferritin that serves as an iron storage protein, and nuclear receptor coactivator 4 (NCOA4), which forms a complex with FTH1 and transfers it to the autolysosome for degradation, is decreased. This indicates that the iron contained in ferritin is released into the intracellular space, leading to oxidative stress [49]. Additionally, after MIRI, the downregulation of USP22 induces the destabilization of SIRT1 via its deubiquitinase activity, leading to p53 upregulation accompanied by a decrease in SLC7A11 expression levels. SLC7A11 is an important ferroptosis suppressor; thus, ferroptosis is induced in cardiomyocytes through the USP22/SIRT1/p53/SLC7A11 pathway [52]. Furthermore, N-acetyltransferase 10 (NAT10), which was previously reported to contribute to apoptosis in MI, is also involved in ferroptosis induction through the regulation of SLC7A11. NAT10 acts as an RNA acetyltransferase and can catalyze the N4-acetylcytidine (ac4C) acetylation of mRNA, increasing its stability and translational efficiency. MYB-binding protein 1A (Mybbp1a) is particularly the target of ac4C by NAT10; therefore, the stability of Mybbp1a mRNA is increased. Mybbp1a recruits p300 and acetylates p53 to enhance p53 transcriptional activation. Consequently, acetylated p53 binds to the SLC7A11 gene promoter and suppresses its expression [53]. Another important pathway recently reported is the Alox15/15-HpETE/Pgc1α axis. The lipoxygenase (LOX) family is known as lipid-peroxidizing enzymes that catalyze the oxygenation of polyunsaturated fatty acids. 15-lipoxygenase (Alox15) produces 15-hydroxyeicosatetraenoic acids (15-HpETE) from arachidonic acid (AA), and the accumulation of 15-HpETE directly induces ferroptosis in cardiomyocytes through mitochondrial dysfunction via the degradation of proliferator-activated receptor gamma coactivator 1-alpha (Pgc1α), an important transcriptional cofactor for regulating mitochondrial quality [54].

### 3.4. Pyroptosis

As well as necroptosis and ferroptosis, pyroptosis has also been recognized as a cause of cardiomyocyte loss in MIRI. A recent study revealed that the expression levels of caspase-11, GSDMD, and their cleaved forms were elevated in both in vivo and in vitro MIRI models. The specific deletion of GSDMD in cardiomyocytes protected the heart from MIRI [55]. The key point in pyroptosis induction is the activation of the inflammasome, which produces active caspase-1. Oxidative stress via ROS production leads to the recruitment of MyD88 to Toll-like receptor 4 (TLR4), and activated NF-kB signaling, including p65 and IκBα downstream of MyD88, causes an increase in the expression of inflammasome components NLRP3 and ASC [56]. This NLRP3/ASC/caspase-1 axis activation is also induced by calpain-mediated ER stress, an increase in mammalian target of rapamycin complex 1 (mTORC1), and the silencing of forkhead box O3 (FoxO3) via the upregulation of miR-29b and miR-149 [57,58,59,60]. Intracellular Ca^2+^ overload in MIRI is considered one of the key factors causing not only apoptosis but also pyroptosis via the NLRP3/ASC/caspase-1 pathway. IRI raises the expression of inositol 1,4,5-trisphosphate receptor (IP3R1), which directly regulates calcium transport from the ER to mitochondria, whereas endoplasmic reticulum resident protein 44 (ERP44), a modulator of IP3R1 activity, declines. Consequently, mitochondrial Ca^2+^ overload is induced in MIRI, triggering inflammasome activation. The overexpression of ERP44 or silencing of IP3R1 can protect the heart from pyroptosis in IRI through a decrease in intracellular Ca^2+^ levels [61]. In addition, ubiquitination and deubiquitination play important roles in pyroptosis. Tripartite motif 16 (TRIM16), an E3 ubiquitin ligase in the TRIM family, is reported to polyubiquitinate NLRP3 through direct interaction with NLRP3, leading to the degradation of NLRP3 and suppression of pyroptosis in MIRI [62]. The other E3 ubiquitin ligase, membrane-associated RING finger protein 2 (MARCH2), also has a protective function in MIRI. MARCH2 interacts with phosphoglycerate mutase family member 5 (PGAM5) and ubiquitinates its K48 residue for degradation. PGAM5 acts as a scaffold co-condensing with mitochondrial antiviral-signaling protein (MAVS) on mitochondria, which is an adaptor protein for the recruitment and activation of the NLRP3 inflammasome. Therefore, the ubiquitination and degradation of PGAM5 via MARCH2 are important for inhibiting the NLRP3 inflammasome and pyroptosis [63]. Furthermore, KAT5 increases the expression of E3 ubiquitin ligase STUB1 through histone acetylation at the STUB1 promoter, leading to the enhancement of the ubiquitination and degradation of LATS2 via direct binding to STUB1. In vitro MIRI models have shown that LATS2 depletion increases the viability of cardiomyocytes, whereas its overexpression promotes pyroptosis through the downregulation of YAP and β-catenin. Thus, KAT5 protects cardiomyocytes from pyroptosis through modulation of the STUB1/LATS2/YAP/β-catenin pathway [64]. On the contrary, the deubiquitination of tumor necrosis factor receptor-associated factor 3 (TRAF3) via USP11 stabilizes TRAF3 expression and exacerbates cardiomyocyte pyroptosis [65].

## 4. Promising Therapies from the Aspect of Programmed Cell Death

Currently, numerous treatment options have been developed for acute coronary syndromes (ACSs), including MIRI. This section focuses on some promising therapies and describes their potential mechanisms in terms of cell death inhibition.

### 4.1. Sodium–Glucose Cotransporter-2 Inhibitors

Sodium–glucose cotransporter-2 (SGLT2) inhibitors, such as empagliflozin, canagliflozin, and dapagliflozin, were originally developed as a new class of antidiabetic agents. However, it has been elucidated that SGLT2 inhibitors also have cardioprotective and renoprotective effects. While SGLT2 inhibitors have demonstrated potential benefits in clinical trials, in the setting of ACS, some reports indicate no clear benefits or even a risk of future heart failure and mortality. Therefore, further clinical trials are still required [66]. Regarding the mechanisms, empagliflozin treatment improved left ventricular fractional shortening and reduced infarct size through the upregulation of STAT3 expression. Alongside STAT3 upregulation, malondialdehyde (MDA) levels, inducible nitric oxide synthetase (iNOS), and interleukin-6 expression were decreased, indicating that the antioxidant and anti-inflammatory roles of STAT3 were activated by empagliflozin treatment [67]. Additionally, canagliflozin reduced infarct size, serum troponin-T levels, apoptotic markers (Bax/Bcl-2 ratio), and 4-hydroxynonenal (HNE) levels through the activation of phosphorylated AMPK and Akt [68]. The Akt pathway is strongly related to cell survival and apoptosis induction, as described above. MDA and 4-HNE are produced through the degradation of lipid peroxides, making them reliable oxidative stress markers and indicators of ferroptosis. Indeed, dapagliflozin was reported to inhibit ferroptosis by suppressing MAPK signaling [69]. Therefore, the inhibition of apoptosis and ferroptosis is thought to contribute to the therapeutic effects of SGLT2 inhibitors. However, AMPK and Akt signaling pathways were not affected by empagliflozin administration, suggesting that the effects on cell death inhibition may differ among SGLT2 inhibitors.

### 4.2. FDY-5301

FDY-5301 is an anti-peroxidant drug based on sodium iodide and optimized in humans for intravenous administration. In MIRI mice models, sodium iodide injection 5 min before reperfusion significantly reduced infarct size, neutrophil infiltration, and serum cTnI levels and improved heart function [70]. In addition, FDY-5301 has already been tested in phase 1 and phase 2a clinical trials. Clinical trial outcomes indicated the safety of FDY-5301 administration, feasibility in emergency settings, and a fast-acting property that increases iodine levels in the blood. Although reductions in infarct size and serum troponin I levels did not reach statistical significance, there was a trend toward reduced infarct size, and the levels of other serum markers, such as myeloperoxidase (MPO), matrix metalloproteinase-2 (MMP-2), and N-terminal pro-brain natriuretic peptide (NT-proBNP), were significantly decreased. Left ventricular function also improved compared to the placebo group. Based on these outcomes, a phase 3 trial is currently underway and is expected to be completed in 2024 [71,72].

On the other hand, the therapeutic mechanism is still unclear. However, iodine is known to act as a catalyst for converting hydrogen peroxide into water and oxygen, potentially protecting the heart from ROS-mediated damage. Oxidative stress induced by ROS can trigger multiple cell death pathways, as described earlier. Therefore, sodium iodide treatment may suppress apoptosis, necroptosis, ferroptosis, and pyroptosis. Another hypothesis proposed by the authors involves thyroid hormones. High blood iodine concentrations reduce thyroid hormone synthesis and secretion due to the Wolff–Chaikoff effect, leading to diminished cardiac metabolism related to reperfusion injury. However, this hypothesis contradicts the consensus on the benefits of thyroid hormone supplementation for cardioprotection. Therefore, further investigation is required [73].

### 4.3. Shuangshen Ningxin Formula

Shuangshen Ningxin Formula (SSNX) is a traditional Chinese medicine composed of three herbs, namely Renshen, Danshen, and Yan Husuo. Its therapeutic effects were tested in a rat MIRI model, demonstrating reductions in infarct size, microvascular damage, and serum cardiac damage markers, along with improvements in cardiac function. Although SSNX has not been assessed in clinical trials for MIRI, its cardioprotective effects are inferred from studies showing its effectiveness in treating stable angina pectoris [74].

Regarding the molecular mechanisms, SSNX reduces activated partial thromboplastin time (APTT), thrombin time (TT), and prothrombin time (PT), which reflect coagulation status. Additionally, SSNX modulates endothelin-1 (ET-1) and endothelial nitric oxide synthase (eNOS), key regulators of vasoconstriction and vasodilation, toward vasodilation. Furthermore, SSNX inhibits the expression of nuclear receptor subfamily 4 group A member 1 (NR4A1), which can induce mitochondrial fission and dysfunction. By doing so, SSNX maintains mitochondrial membrane potential and suppresses mitochondria-mediated apoptosis in endothelial cells. Therefore, SSNX protects the heart from MIRI by improving cardiac microvasculature. Simultaneously, apoptosis in cardiomyocytes was also reduced in a rat MIRI model, though specific mechanisms were not detailed. NR4A1 might be similarly involved in cardiomyocytes. Moreover, SSNX potentially inhibits necroptosis, ferroptosis, and pyroptosis, since NR4A1 is associated with multiple cell death pathways. The details are described in the next section [75].

### 4.4. C-Reactive Protein Apheresis

C-reactive protein (CRP) is commonly used as an inflammation marker in blood tests. However, it also plays an important role in immune responses. In dead or unhealthy cells (e.g., damaged, energy-depleted, or hypoxic cells), phosphatidylcholine (PC) in the outer plasma membrane is partially hydrolyzed to lysophosphatidylcholine (LPC) by secretory phospholipase A2 type IIa (sPLA2 IIa). CRP recognizes and directly binds to LPC, recruiting and activating the classical complement pathway. Consequently, phagocytes such as macrophages eliminate CRP-labeled cells and produce interleukin-6, which further stimulates CRP production in hepatocytes.

Under normal circumstances, CRP only labels dead or dying cells. However, excessive CRP levels can also target viable cells, exacerbating inflammation and tissue damage [3]. Therefore, CRP removal through apheresis is an effective therapeutic option for various diseases, demonstrating efficacy in acute MI and COVID-19 patients [76,77]. Although no histological or cell death analysis was conducted in these reports, it is hypothesized that suppressing the inflammatory response, including TNFα secretion, might inhibit extrinsic apoptosis and necroptosis, which depend on death receptor activation by TNFα.

## 5. Therapeutic Targets for Myocardial Ischemia–Reperfusion Injury via Cell Death Inhibition

Some therapeutic candidates for MIRI, which have been reported to have anti-apoptotic roles among their effects, have been tested in clinical trials; however, they could not provide improved clinical outcomes despite the suppression of apoptosis [6,7]. One of the reasons for this failure is thought to be the involvement of various types of cell death in the progression of MIRI, as described above. In fact, the suppression of two types of cell death has been shown to protect the heart from MIRI more effectively [78,79,80].

Pang et al. discovered a novel TNNI3K inhibitor that demonstrated cardioprotective effects in a rat MIRI model through the inhibition of pyroptosis and apoptosis [78]. Tu et al. tested ponatinib and deferoxamine to inhibit necroptosis and ferroptosis in vitro and in a rat MIRI model, revealing that this combination therapy could more effectively reduce MIRI compared to the administration of ponatinib or deferoxamine only [79]. Koshinuma et al. attempted to suppress necroptosis and apoptosis through the combination injection of necrostatin-1 and Z-VAD-FMK, finding that MIRI in isolated pig hearts was significantly reduced compared to individual inhibition of apoptosis or necroptosis [80]. Therefore, the development of medications and therapeutic options to suppress multiple cell death pathways will be required for more effective therapy. Suggestions for some candidates for a multiple cell death inhibition strategy is shown in the table below (Table 1).

### 5.1. PANoptosis

Recently, the new concept of PANoptosis was defined as an inflammatory programmed cell death triggered by the PANoptosome complex, which regulates pyroptosis, apoptosis, and/or necroptosis simultaneously. The PANoptosome complex consists of several important factors involved in pyroptosis, apoptosis, and necroptosis, such as FADD, caspase-1, caspase-8, ASC, and NLRP3. Z-DNA binding protein 1 (ZBP1) and transforming growth factor beta-activated kinase 1 (TAK1) have been reported as master regulators of PANoptosis. The activation of ZBP1 facilitates the assembly of the PANoptosome complex, while the deletion or suppression of TAK1 enhances PANoptosis [83]. In a rat MIRI model, it was revealed that the expression level of ZBP1 was significantly increased, and the therapeutic drug penehyclidine hydrochloride (PHC) attenuated infarct size, pathological damage in myocardial tissue, and the level of cardiac damage markers in serum through the inhibition of PANoptosis by reducing ZBP1 expression [81].

On the other hand, there have been no reports indicating the involvement of TAK1-mediated PANoptosis in MIRI; however, some articles mentioned the role of TAK1 in individual forms of cell death in MI and MIRI. TAK1 inhibition mitigated MIRI through the reduction in apoptosis, ROS, and ER stress [84]. Additionally, pyroptosis in MI is also suppressed by inhibiting the TAK1/JNK signaling pathway [85]. In contrast, the enhancement of TAK1 phosphorylation leads to the suppression of necroptosis in MIRI [86]. Although further investigation is required due to the limited number of articles addressing TAK1 function in MIRI and the presence of controversial papers, TAK1 has the potential to be a therapeutic target alongside ZBP1.

Besides ZBP1 and TAK1, Piezo1 has recently been reported as a new regulatory factor in the PANoptosis pathway. Piezo1, a mechanosensitive ion channel, is expressed in various types of cells, including cardiomyocytes, and is known to convert mechanical stimuli into intracellular ROS and calcium signaling. In MIRI, PANoptosis-related factors were upregulated in accordance with an increase in the expression level of Piezo1. In particular, Piezo1 directly interacted with caspase-8, leading to the activation of the PANoptosome complex, independent of calcium signaling. Additionally, the pharmacological inhibition of Piezo1 using GsMTx4 significantly reduced Piezo1 expression and heart damage through PANoptosis inhibition in both in vitro and in vivo MIRI models, while the activation of Piezo1 due to Yoda1 treatment aggravated cell viability by enhancing PANoptosis in vitro [82]. Therefore, the regulatory molecules of PANoptosis have the potential to reduce cardiac tissue damage caused by ischemia/reperfusion.

### 5.2. Nrf2

Nuclear factor erythroid 2-related factor 2 (Nrf2), an important transcription factor for antioxidant genes, is related to apoptosis, pyroptosis, and ferroptosis. Regarding apoptosis, the upregulation of the Nrf2/HO-1 signal via PERK overexpression and pharmacological treatment protected mouse hearts from MIRI by suppressing ER stress and oxidative stress [87,88]. The activation of the Nrf2/HO-1 pathway in MIRI models suppressed pyroptosis by decreasing oxidative stress and the expression level of the NLRP3 inflammasome complex [89]. Additionally, ferroptosis, oxidative stress, and inflammation were reduced through the Nrf2/HO-1 axis by regulating SLC7A11 and GPx4 [90]. A recent study showed that the inhibition of MALT1 reduces ferroptosis in rat MIRI and in vitro IRI models by enhancing the Nrf2/SLC7A11 pathway [91]. Therefore, Nrf2 can be targeted to inhibit three types of cell death.

### 5.3. SIRT1

*Sirtuin-1* (*SIRT1*) is a NAD-dependent deacetylase well-known as an important gene for resistance to apoptosis, inflammation, and oxidative stress, while its involvement in other types of cell death has also been indicated recently. In MIRI, an increase in SIRT1 level via pharmacological treatment prevents apoptosis, inflammation, oxidative stress, and excessive autophagy through the activation of the mTOR pathway [92]. Similarly, another reagent suppresses apoptosis through the SIRT1/Foxo1 pathway [93]. Regarding pyroptosis, IRI increases the expression of miR-29a, which inhibits the expression of SIRT1. The upregulation of SIRT1 by miR-29a knockdown suppresses oxidative stress and NLRP3-mediated pyroptosis [94]. As described above, SIRT1 can also regulate ferroptosis through the p53/SLC7A11 axis [52]. While the enhancement of SIRT1 expression has advantages in MIRI treatment, extreme overexpression of SIRT1 has been reported to lead to disadvantages such as hypertrophy and cardiac dysfunction. Therefore, tight control of SIRT1 expression is required to apply SIRT1-mediated treatment for MIRI [95].

### 5.4. NR4A1

Nuclear receptor subfamily 4 group A member 1 (NR4A1), which is included in the steroid–thyroid hormone receptor superfamily, has a regulatory function in multiple pathways, such as the inflammatory response, glucose and lipid metabolism, adipogenesis, and tumorigenesis. Additionally, it has been reported that NR4A1 plays a role as an apoptosis regulator. In pancreatic ß-cells, NR4A1 shows a protective function through the upregulation of anti-apoptotic genes (*Survivin*, *WT1*, *Bcl-2*) and the antioxidant gene *superoxide dismutase 1* (*SOD1*). On the other hand, the translocation of NR4A1 from the nucleus to mitochondria induces cytochrome C release and apoptosis in cancer cells. Furthermore, a decrease in mitochondrial membrane potential via BAX translocation to mitochondria is also controlled by NR4A1 when RAW 264.7 cells, a macrophage cell line, are treated with Simvastatin. Therefore, NR4A1 exhibits both pro-apoptotic and anti-apoptotic functions depending on cell types [97]. However, a recent report indicated that NR4A1 can suppress not only apoptosis but also necroptosis, ferroptosis, and pyroptosis in MIRI. Ischemia–reperfusion injury causes an increase in NR4A1 expression, accompanied by changes in mitochondrial morphology and structure, leading to oxidative stress and mitochondrial disorder. This mitochondrial dysfunction is triggered by the promotion of mitochondrial fission and the suppression of mitophagy. Drp1 and mitochondrial fission factor (Mff), which are important factors in mitochondrial fission, are significantly increased after MIRI, resulting in overactivated mitochondrial fragmentation. Regarding the mechanism of mitophagy, FUN14 domain 1 (FUNDC1) plays a crucial role. FUNDC1 was recently identified as a mitochondrial receptor that is required for mitochondrial quality control via mediating mitochondrial dynamics. However, FUNDC1 expression is suppressed after MIRI, causing the dysregulation of mitophagy. Mitochondrial dysfunction is strongly related to multiple cell death pathways, including apoptosis, necroptosis, pyroptosis, and ferroptosis; therefore, gene knockout of NR4A1 can protect the heart from MIRI by inhibiting multiple types of cell death through the suppression of Mff-mediated mitochondrial fission and the improvement of FUNDC1-mediated mitophagy. Although further investigation is still required, as it is unclear how NR4A1 regulates Mff and FUNDC1 expression, NR4A1 has the potential to be a therapeutic target for effectively suppressing multiple types of cell death in MIRI [96].

## 6. Potential Risks of Cell Death Inhibition

Cell death is a biological mechanism that eliminates abnormal cells, which pose a potential threat to organisms due to damage or mutations. Therefore, inhibiting excessive cell death can be beneficial in mitigating disease states. However, the long-term suppression of cell death carries potential risks that need to be carefully considered.

The most predictable risk is tumorigenesis. Mutations and malfunctions in the *TP53* gene, which is a well-known tumor suppressor, are significant contributors to tumorigenesis, as *TP53* plays a crucial role in regulating apoptosis and ferroptosis in response to cellular damage such as DNA damage [98]. Additionally, certain cancer types, including chronic lymphocytic leukemia, colon cancer, and breast cancer, exhibit low sensitivity to necroptosis due to reduced RIPK3 expression, which is associated with poor prognosis [99,100,101]. Therefore, the inhibition of cell death may facilitate cancer progression, and the use of cell death inhibitors should generally be contraindicated in cancer patients.

Another risk associated with inhibiting cell death is the potential development of autoimmune diseases. For instance, autoimmune lymphoproliferative syndrome (ALPS), a rare inherited disorder, arises from mutations in the Fas signaling pathway. Normally, autoreactive T cells are eliminated through Fas-mediated apoptosis. However, defects in this pathway caused by mutations permit the expansion and proliferation of autoreactive T cells, ultimately leading to autoimmune disease [102]. Thus, the long-term inhibition of cell death could create conditions similar to those observed in ALPS, promoting the survival and activity of autoreactive immune cells. Although the necroptosis pathway might act as an alternative mechanism of cell death when apoptosis alone is suppressed, as described in Section 2.2, the potential risks associated with the simultaneous inhibition of multiple cell death pathways should not be overlooked when applying such strategies to patients.

To avoid potential risks via cell death inhibition, the specific delivery of inhibitors is very important. Recently, nanoparticles consisting of poly(lactic-co-glycolic) acid (PLGA) or the combination of poly(ethylene glycol) (PEG) and poly(propylene sulfide) (PPS) were developed for ischemia–reperfusion injury. Their encapsulation is controlled by pH change or the existence of ROS; thus, components encapsulated in nanoparticles were selectively released in ischemic areas [103,104]. Furthermore, numerous reports using biomaterials such as nanoparticles, liposomes, and nanocomplexes have been published [105]. Selective delivery enables us to not only avoid off-target effects but also achieve high therapeutic efficacy due to improving bioavailability. Therefore, further development is expected.

## 7. Future Direction

Cell death pathways have been investigated enthusiastically for decades, and it has been demonstrated that cell death inhibition can be an effective therapeutic option. One important issue is the uncertainty of the cell fate after removing cell death inhibition. Even though cell death pathways are inhibited, cellular damage is possibly retained after completing treatment. In the case that cells are unable to recover from damage, it is anticipated that cell death pathways are eventually reactivated or cellular senescence is induced. In particular, it is known that senescent cells show a senescence-associated secretory phenotype (SASP) and release proteins, cytokines, and other factors, leading to inflammation and tumorigenesis. In the cardiac context, even though the incident rate of cardiac tumors is vanishingly low, chronic inflammation can be a risk factor for cardiac dysfunction. Therefore, the study of the long-term effect of cell death inhibition needs to be evaluated as a preclinical priority.

## 8. Conclusions

In addition to numerous reports on the contribution of apoptosis to homeostasis, development, and diseases over the years, the number of articles addressing other forms of PCD has dramatically increased in recent decades. This trend reflects the growing interest in cell death research, driven by the discovery of molecular pathways involved in PCDs. Recent studies have uncovered the complexity of PCD induction in the onset and progression of diseases, demonstrating that various types of cell death contribute simultaneously to MIRI. In this review, we highlighted the fundamental pathways of the four major types of PCDs, which are apoptosis, necroptosis, ferroptosis, and pyroptosis. We also reviewed the specific molecular mechanisms of these PCDs in MIRI and presented potential therapeutic targets aimed at inhibiting multiple cell death pathways. Theoretically, inhibiting multiple cell death pathways should be more effective than targeting a single cell death pathway. The combination of two inhibitors or the discovery of novel inhibitors that simultaneously affect different molecular pathways could more effectively reduce tissue damage caused by ischemia–reperfusion. Therefore, further investigation of PCD pathways and the development of therapeutic options to suppress multiple types of cell death are essential for preventing MIRI, a life-threatening condition.

## Figures and Tables

**Figure 1 ijms-25-13492-f001:**
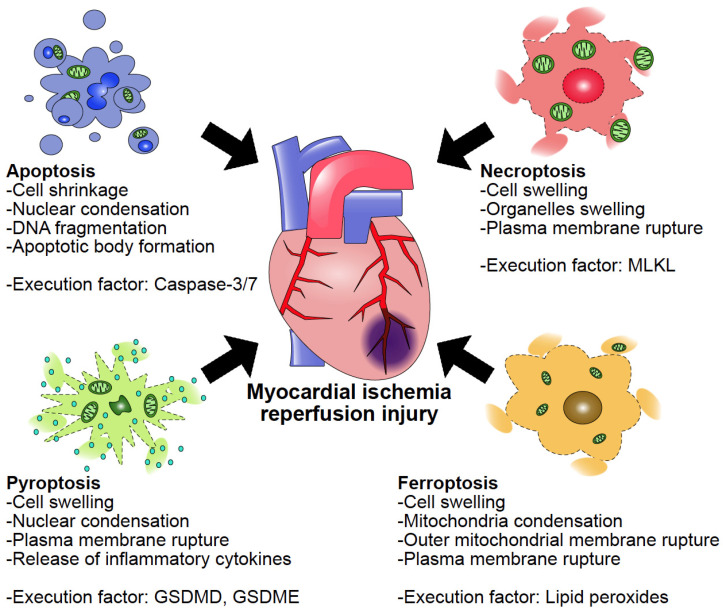
The involvement of multiple cell death pathways in cardiomyocyte loss in MIRI. The blockage of the coronary artery by cholesterol plaque and blood clots causes hypoxia and damages cardiomyocytes. In addition, restoring blood supply via revascularization results in a further loss of cardiomyocytes through massive production of reactive oxygen species, causing reperfusion injury. In this process, various types of cell death are induced and involved in cardiomyocyte loss.

**Table 1 ijms-25-13492-t001:** Candidate genes having the potential to inhibit multiple cell death pathways simultaneously in MIRI.

Gene	Target Cell Death Form	Related Mechanisms	References
ZBP1	PANoptosis (Apoptosis, Necroptosis and Pyroptosis)	PANoptosis: activation of ZBP1 enhances the interaction with RIPK3 and caspase-8, which are the components of PANoptosome complex, leading to the induction of PANoptosis.	[81]
Piezo1	PANoptosis (Apoptosis, Necroptosis and Pyroptosis)	PANoptosis: Piezo1 directly interacts with caspase-8, causing the activation of PANotpsome.	[82]
TAK1	Apoptosis Necroptosis Pyroptosis	Apoptosis: TAK1 inhibition leads to oxidative stress and ER stress reduction. Necroptosis: an increase in TAK1 phosphorylation enhances TAK1 binding to RIPK1 to inhibiting the RIPK1/RIPK3 complex. Pyroptosis: blocking the TAK1/JNK pathway suppresses the activation of the NLRP3 inflammasome complex (in MI, not MIRI).	[83,84,85,86]
Nrf2	Apoptosis Pyroptosis Ferroptosis	Apoptosis: activation of the antioxidant Nrf2/OH-1 pathway reduces oxidative stress and ER stress. Pyroptosis: reduction in oxidative stress leads to the downregulation of the NLRP3 inflammasome complex. Ferroptosis: activation of the Nrf2/OH-1 pathway upregulates ferroptosis suppressor SLC7A11 and GPx4.	[87,88,89,90,91]
SIRT1	Apoptosis Pyroptosis Ferroptosis	Apoptosis: activation of the mTOR pathway and suppression of the acetylated Foxo1 level. Pyroptosis: suppression of oxidative stress leads to the downregulation of the NLRP3 inflammasome complex. Ferroptosis: upregulation of the SLC7A11 expression level through suppression of the p53 expression level.	[52,92,93,94,95]
NR4A1	Apoptosis Necroptosis Ferroptosis Pyroptosis	Suppression of NR4A1 reduces mitochondrial dysfunction through the improvement of mitochondrial homeostasis, leading to the inhibition of multiple cell death pathways simultaneously.	[96]

## Data Availability

No new data were created or analyzed in this study. Data sharing is not applicable to this article.

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
