# Peer review of "Mechanisms and Therapeutic Potential of Multiple Forms of Cell Death in Myocardial Ischemia–Reperfusion Injury"

_ijms, 2024, doi:10.3390/ijms252413492_

Round 1
Reviewer 1 Report
Comments and Suggestions for Authors
The manuscript provides an in-depth review of various forms of programmed cell death (PCD) and their roles in MIRI, focusing on apoptosis, necroptosis, ferroptosis, and pyroptosis. This is a valuable topic, as targeting cell death pathways holds promise for mitigating cardiac damage during reperfusion therapy. The review also discusses several therapeutic targets, highlighting inhibitors of cell death pathways. The following limitations should be resolved:
The section on therapeutic targets for MIRI would benefit significantly from expanding on the emerging role of sodium-glucose cotransporter 2 (SGLT2) inhibitors in this setting. While traditionally used in diabetes management, recent research shows that SGLT2 inhibitors may also provide cardioprotective effects in acute settings like MIRI by reducing oxidative stress, inflammation, and possibly enhancing mitochondrial function. Including a discussion on their potential as therapeutic agents in ischemia-reperfusion injury would make this review more comprehensive and up-to-date, given the increasing focus on SGLT2 inhibitors in cardiovascular protection. Please refer to DOI: 10.1016/j.clinthera.2024.06.010
The manuscript suggests that simultaneous inhibition of multiple cell death pathways could be effective in reducing myocardial injury. However, clearer guidance on the specific benefits and challenges of this approach would add depth. For instance, a brief discussion on the risk of interfering with necessary cellular processes or potential off-target effects in clinical scenarios would make the proposed multi-target strategy more balanced.
Certain sections, such as those on emerging therapies, could benefit from additional detail on the molecular mechanisms by which these therapies may interact with cell death pathways. For instance, elucidating how specific targets modulate oxidative stress, mitochondrial stability, or inflammatory cascades would provide a more complete picture of how these treatments might mitigate MIRI.
While various agents targeting apoptosis, necroptosis, ferroptosis, and pyroptosis are discussed, the manuscript could be strengthened by adding a comparative perspective. For example, highlighting studies that compare the effectiveness of targeting single versus multiple cell death pathways would add clarity to the potential benefits of a multi-target approach.
Adding a section or paragraph suggesting future research directions could make the review more impactful. This might include calls for additional studies to elucidate mechanisms of PCD in MIRI, to develop combination therapies that minimize off-target effects, or to explore patient-specific factors (e.g., genetic predispositions) that may influence treatment efficacy.
Author Response
|
Comments 1: The section on therapeutic targets for MIRI would benefit significantly from expanding on the emerging role of sodium-glucose cotransporter 2 (SGLT2) inhibitors in this setting. While traditionally used in diabetes management, recent research shows that SGLT2 inhibitors may also provide cardioprotective effects in acute settings like MIRI by reducing oxidative stress, inflammation, and possibly enhancing mitochondrial function. Including a discussion on their potential as therapeutic agents in ischemia-reperfusion injury would make this review more comprehensive and up-to-date, given the increasing focus on SGLT2 inhibitors in cardiovascular protection. Please refer to DOI: 10.1016/j.clinthera.2024.06.010.
|
|
Response 1: Thank you for suggesting this target. Following this comment, we have described the cardioprotective effect of SGLT2 inhibitors and how SGLT2 inhibitors can suppress cell death pathways in MIRI. Because SGLT2 inhibitors have been tested in clinical trials, we have added the new section for emerging therapies and illustrated SGLT2 inhibitors with other therapeutic options in it. This change can be found at page 8 in the section 4, promising therapies from the aspect of programmed cell death.
|
|
Comments 2: The manuscript suggests that simultaneous inhibition of multiple cell death pathways could be effective in reducing myocardial injury. However, clearer guidance on the specific benefits and challenges of this approach would add depth. For instance, a brief discussion on the risk of interfering with necessary cellular processes or potential off-target effects in clinical scenarios would make the proposed multi-target strategy more balanced. |
|
Response 2: We agree completely. We have, accordingly, added the new section to describe potential risks of cell death inhibition. It was difficult to refer the articles obviously showing the disadvantages of cell death inhibition. However, we have given tumorigenesis and autoimmune diseases as examples because mutations in cell death-related genes perturbs cell death induction in both cases and long-term cell death inhibition may be able to create the similar situation. This change can be found at page 12 in the section 6, potential risks of cell death inhibition.
|
|
Comments 3: Certain sections, such as those on emerging therapies, could benefit from additional detail on the molecular mechanisms by which these therapies may interact with cell death pathways. For instance, elucidating how specific targets modulate oxidative stress, mitochondrial stability, or inflammatory cascades would provide a more complete picture of how these treatments might mitigate MIRI. |
|
Response 3: We fully agree with your comment. Accordingly, we have added a new section to discuss promising therapies, including SGLT2 inhibitors, which have recently shown positive outcomes in clinical trials. While the involvement of cell death and its molecular pathways has been demonstrated for some therapies, others have primarily shown effects such as the reduction of infarct size, lower serum cardiac damage markers, and improved cardiac function. Therefore, we have described the potential mechanisms of these therapies in inhibiting cell death pathways, drawing on findings from previous studies. We hope this new section provides a clearer perspective on the role of cell death pathways in emerging therapeutic strategies. This change can be found at page 8 in the section 4, promising therapies from the aspect of programmed cell death.
|
|
Comments 4: While various agents targeting apoptosis, necroptosis, ferroptosis, and pyroptosis are discussed, the manuscript could be strengthened by adding a comparative perspective. For example, highlighting studies that compare the effectiveness of targeting single versus multiple cell death pathways would add clarity to the potential benefits of a multi-target approach. |
|
Response 4: Thank you for pointing this out. We aimed to demonstrate that the multi-target approach had already been compared to targeting a single type of cell death in the references, highlighting the greater effectiveness of inhibiting multiple cell death pathways. However, the original description may not have made this point sufficiently clear. Therefore, we revised some sentences to enhance clarity and ensure ease of understanding. These changes can be found on pages 9 and 10, lines 425 and 428, in section 5, therapeutic targets for myocardial ischemia-reperfusion injury via cell death inhibition.
|
|
Comments 5: Adding a section or paragraph suggesting future research directions could make the review more impactful. This might include calls for additional studies to elucidate mechanisms of PCD in MIRI, to develop combination therapies that minimize off-target effects, or to explore patient-specific factors (e.g., genetic predispositions) that may influence treatment efficacy.
|
|
Response 5: We completely agree with your comment. In response, we have elaborated on how biomaterials, such as nanoparticles, can be utilized for target-specific delivery, not only to minimize off-target effects but also to enhance bioavailability. This change can be found on page 12 in section 6, potential Risks of Cell Death Inhibition. Additionally, we have introduced a new section addressing future research directions. While the therapeutic efficacy of cell death inhibition has been extensively studied, the fate of cells following treatment remains unclear. It is crucial to determine whether cells can recover from the damage or develop abnormalities after the inhibition of cell death pathways, as this may significantly impact prognosis. Therefore, we have emphasized the importance of investigating the long-term effects of cell death inhibition in the review. This change can be found at page 13 in the section 7, future direction.
|

Reviewer 2 Report
Comments and Suggestions for Authors
The authors describe the different types of cell death well.
Line 32: What kind of damage takes place under oxygen deprivation? - This has already been described in:
doi: 10.3389/fimmu.2021.630430
doi.org/10.3390/biomedicines11082279
doi.org/10.3390/jcm11216446
doi.org/10.3390/jcm11071771
doi: 10.3389/fcvm.2021.591714
https://doi.org/10.1007/s00063-022-00911-x
Lines 60-62: The aforementioned publications point to a different conclusion. I am surprised that the numerous reports are not known. In Ries et al (Figure 3, apheresis) one sees CMRs after STEMI from patients after CRP depletion in which no infarct area is seen. How can this be if the cell death fate of the cardiomyocytes is determined by the ischemia and MIRI? Cell death blockers were not necessary for this. If you also look at the publications on CRP apheresis in severe COVID-19, you can see the same effect in the patients' lungs. Due to the easier imaging, you can even see that the lungs already showing signs of pneumonia look much better again after the end of treatment, which clearly illustrates that no form of cell death described here has been induced in any way.
doi: 10.3389/fimmu.2021.708101
https://doi.org/10.3390/jcm11071956
I am really sorry to have to say this, but the cell death argument does not reflect the current state of knowledge.
Author Response
|
Comments 1: Line 32: What kind of damage takes place under oxygen deprivation? - This has already been described in: |
|
Response 1: Thank you for indicating the interesting mechanisms in exacerbation of ischemic injury through CRP. We partially agree with this comment and have cited some of articles you referred at page 1, line 32. In references you indicated and related articles, the therapeutic effect of CRP apheresis was obviously demonstrated, and apoptotic cell clearance via macrophages was enhanced through CRP-complement pathway (PMID: 9858517). However, there is no evidence showing energy-depleted or ischemic cells are eliminated by phagocytes while cells still survive. Thus, the molecular mechanisms proposed in the references appear to remain conceptual. Considering these points, we agree that CRP exacerbates the inflammatory response and tissue damage under oxygen deprivation.
|
|
Comments 2: Lines 60-62: The aforementioned publications point to a different conclusion. I am surprised that the numerous reports are not known. In Ries et al (Figure 3, apheresis) one sees CMRs after STEMI from patients after CRP depletion in which no infarct area is seen. How can this be if the cell death fate of the cardiomyocytes is determined by the ischemia and MIRI? Cell death blockers were not necessary for this. If you also look at the publications on CRP apheresis in severe COVID-19, you can see the same effect in the patients' lungs. Due to the easier imaging, you can even see that the lungs already showing signs of pneumonia look much better again after the end of treatment, which clearly illustrates that no form of cell death described here has been induced in any way. doi: 10.3389/fimmu.2021.708101 |
|
Response 2: Thank you for showing this promising therapeutic option. These publications showed the amazing therapeutic potential of CRP depletion in STEMI and COVID-19 patients. However, we disagree because your comment does not fit with the premise of our review for the following reasons. 1: Upregulation of CRP is supposed to require other factors such as cellular damage and/or cell death to initiate inflammatory responses. Our review focused on the roles of cell death in not only progression but also the onset of disease states which can trigger inflammatory responses including upregulation of CRP. 2: A reference you indicated (doi.org/10.3390/jcm11216446) illustrated that CRP apheresis was less effective in STEMI patients who showed the increase-rate of the CRP amount was lower than 0.6. Considering that, CRP appears to be involved in amplifying inflammation and tissue damage rather than initiating them. 3: In the publications you indicated, only X-ray imaging or cardiac magnetic resonance were conducted. It is unclear whether cell death is induced or not before/after CRP apheresis treatment because histological analysis and cell death detection assays were not conducted. 4: If hyperactivity of CRP-mediated phagocytosis is the only cause of tissue damage, the cell death inhibitors are supposed to be ineffective. This is not the case. Therefore, our opinion is that the evidence you indicated cannot completely deny the involvement of cell death.
On the other hand, as described in response 1, we agree that CRP apheresis can suppress the exacerbation of inflammation and tissue damage. In addition, we would like to suggest that extrinsic apoptosis and necroptosis, which are induced by inflammatory cytokines such as TNFa, can be promoted if CRP exaggerates inflammation. Therefore, we described CRP apheresis in the section for promising therapies. This change can be found at page 9, line 395 to 411 in the section 4, promising therapies from the aspect of programmed cell death.
|

Round 2
Reviewer 1 Report
Comments and Suggestions for Authors
I would like to thank the authors for appropriately revising their manuscript.